# ECoFLaP: Efficient Coarse-to-Fine Layer-Wise Pruning for Vision-Language Models

**Yi-Lin Sung    Jaehong Yoon    Mohit Bansal**
Department of Computer Science, UNC Chapel Hill
{ylsung, jhyoon, mbansal}@cs.unc.edu

## Abstract

Large Vision-Language Models (LVLMs) can understand the world comprehensively by integrating rich information from different modalities, achieving remarkable advancements on various multimodal downstream tasks. However, deploying LVLMs is often problematic due to their massive computational/energy costs and carbon consumption. Such issues make it infeasible to adopt conventional *iterative global pruning*, which is costly due to computing the Hessian matrix of the entire large model for sparsification. Alternatively, several studies have recently proposed layer-wise pruning approaches to avoid the expensive computation of global pruning and efficiently compress model weights according to their *importance* within a layer. However, they often suffer from suboptimal model compression due to their lack of a global perspective. To address this limitation in recent efficient pruning methods for large models, we propose *Efficient Coarse-to-Fine Layer-Wise Pruning (ECoFLaP)*, a two-stage *coarse-to-fine* weight pruning approach for LVLMs. We first determine the sparsity ratios of different layers or blocks by leveraging the global importance score, which is efficiently computed based on the zeroth-order approximation of the global model gradients. Then, the model performs local layer-wise unstructured weight pruning based on globally-informed sparsity ratios. We validate our proposed method across various multimodal and unimodal models and datasets, demonstrating significant performance improvements over prevalent pruning techniques in the high-sparsity regime.

## 1 Introduction

Deep learning models have been increasingly growing in size (Radford et al., 2018; Brown et al., 2020; Liu et al., 2023; Zhu et al., 2023) in order to have a sufficient capacity to learn challenging tasks in the real world. However, this exorbitant model size requires significant computations and memory to deploy, which limits their applicability in many resource-constrained environments. Model compression, a strategy to reduce the size of neural networks while preserving their capabilities (Dai et al., 2018; Park et al., 2022; Xiao et al., 2023; Fang et al., 2023), has gained popularity in tackling this problem. In order to build lighter, faster, and more interpretable models, various directions have been studied in parallel, including model pruning (Dai et al., 2018; Fang et al., 2023; Frantar & Alistarh, 2023), quantization (Park et al., 2022; Xiao et al., 2023), layer drop (Sajjad et al., 2023), and token merging (Bolya et al., 2022; Bolya & Hoffman, 2023). Among them, in this paper, we study an unstructured model pruning approach that has strong potential to preserve model performance, even with a high compression rate of large vision-language models.

While most model pruning approaches tackle the problem of vision-based tasks (Liang et al., 2022b; Kong et al., 2022; Wei et al., 2023) and a few recent works have studied to reduce the size of language models (Ma et al., 2023; Sun et al., 2023a; Frantar & Alistarh, 2023), efficient pruning for multimodal models like Large Vision-Language Models (LVLMs) (Ye et al., 2023; Li et al., 2023a) have been understudied due to their architectural complexity and the disparities in data characteristics between different modalities (Shi et al., 2023). Unlike unimodal learning methods, network architectures for multimodal learning (Li et al., 2022; 2023c) are often composed of modality-specific sub-modules to capture the knowledge for each modality, which substantially expands the scale of multimodal models.

---

Our project page and code are available at https://ecoflap.github.io/

Moreover, this modularization leads to significant imbalances in the weight/gradient distributions between modules associated with different modalities, making unified pruning difficult.

As conventional iterative pruning (Mallya & Lazebnik, 2018; Molchanov et al., 2019; Wang et al., 2020; Zhang et al., 2022b) demands extensive computations to learn pruning masks by computing the inverse Hessian in large models, layer-wise one-shot pruning (Chen & Zhao, 2018; Frantar & Alistarh, 2023; Sun et al., 2023a) has recently been developed to compress them efficiently. These approaches discard uninformative model weights in a single iteration without computing the expensive second-order Hessian matrix for the entire model architecture, yet often suffer from finding the optimal sparsity ratio per layer, as they resort to intra-layer weight importance without sufficient understanding of global (i.e., inter-layer) weight correlations in the model, hence leading to suboptimal pruning.

To overcome this limitation of layer-wise pruning and build an efficient approach for LVLM compression, we propose *Efficient Coarse-to-Fine Layer-Wise Pruning (ECoFLaP)* that obtains an adaptive sparsity ratio per layer in a single step based on a 'global importance score' (*Coarse*) and then removes parameters that are less critical (to the model's performance), in a layer-wise manner (*Fine*). Note that our proposed method is computationally efficient by leveraging the first-order gradient to obtain a global importance score without Hessian operations, allowing layer-wise pruning to leverage a holistic correlation across the model weights. To further improve the memory efficiency of the global score computation, we introduce the zeroth-order approximate gradient computed by the forward-forward algorithm (Malladi et al., 2023; Hinton, 2022). In the end, our ECoFLaP obtains highly compressed large models in a single shot, with adaptive sparsity in each layer, leveraging global weight importance approximated by memory-efficient forward-forward operations. We note that our proposed pruning method generalizes well not only to vision-language multimodal tasks but also to unimodal tasks, including ImageNet (Vision) and MMLU (NLP).

We extensively validate the efficacy of our approach across a variety of models and datasets, including but not limited to EVA-ViT, FlanT5, LLaMA, BLIP/BLIP-2, CLIP, and on image classification (ImageNet-1k), multitask multiple-choice question answering (MMLU), next token prediction (Wiki-Text), visual reasoning (NLVR$^2$), visual question answering (VQAv2, OK-VQA, GQA), image captioning (NoCaps, COCO Captions), and image-text retrieval (Flickr30k). Our proposed ECoFLaP surpasses iterative global pruning and SoTA layer-wise pruning methods, *SparseGPT* (Frantar & Alistarh, 2023) and *Wanda* (Sun et al., 2023a), with relative improvements up to $5\%$ in average performance across multiple VL tasks. Additionally, our approach outperforms a recent SoTA in pruning vision-language model, UPop (Shi et al., 2023), achieving relative improvements of 1.8% and 2.6% on NLVR$^2$ and COCO captions. ECoFLaP also generalizes well to unimodal models (FlanT5, LLaMA, EVA-ViT), particularly in a higher sparsity regime. It is worth noting that ECoFLaP with the zeroth-order gradient reduces the GPU memory usage by up to $40\%$ compared to baselines using the first-order gradient to estimate the global importance. These results not only underscore the potential of our method but also pave the way for more efficient and effective pruning techniques in the future.

We summarize our contributions as threefold:

• We propose a novel layer-wise pruning method for vision-language models, coined **E**fficient **Co**arse-to-**F**ine **La**yer-Wise **P**runing for Vision-Language Models (**ECoFLaP**), that finds adaptive sparsity per layer by leveraging the global importance score approximated via zeroth-order gradients.

• We validate the proposed method on various backbone structures with diverse vision-language and unimodal tasks, and demonstrate the superiority of our method by consistently outperforming the SoTA pruning baselines for large models on multiple benchmark datasets.

• We provide extensive analyses and ablation studies that help to understand the challenges of unified multimodal model pruning and the strengths of our proposed methods compared to baselines, including visualizations of importance score distribution, dynamic sparsity, loss landscape, etc.

## 2 RELATED WORK

**Model pruning for transformers.** Efficient model inference via pruning (Yoon & Hwang, 2017; Dai et al., 2018; Zhang et al., 2022a; Fang et al., 2023) is one of the long-standing topics in the community and is coming to the fore as transformer-based models grow in size. Model pruning is typically categorized into *structured pruning* and *unstructured pruning*. The former (McCarley et al., 2019; Wang et al., 2019; Kwon et al., 2022) aims to accelerate inference speed and throughput at

the expense of model performance. Unstructured pruning (Sun et al., 2023a; Frantar & Alistarh, 2023), on the other hand, can be advantageous in preserving performance even with high model sparsity given AI acceleration software or sparse matrix computation schemes (Han et al., 2016; Mishra et al., 2021; Das & Ramamoorthy, 2022; NeuralMagic, 2022). Recently, Frantar & Alistarh (2023) suggest a one-shot pruning technique, SparseGPT, for generated pre-trained transformers (GPTs) in an unstructured manner. They newly employ a sparse regression solver that prunes weights at each layer based on row-wise Hessian reconstruction as formulated by a closed-form solution. Wanda (Sun et al., 2023a) proposes a magnitude-based unstructured pruning approach for large language models (LLMs). It promotes layer-wise weight sparsification based on the importance, computed by multiplying weights and input activations. However, these SoTA pruning methods rely on the pre-defined pruning ratio that all layers resort to the same sparsity, restricting the upper bound on model compression. In addition, their methods are tailored to language models without concern for different modalities. On the other hand, our proposed method allows adaptive pruning at each layer without heavy computation of global gradients. Further, to the best of our knowledge, we propose a first unified sparse approximate solver for vision-language multimodal models.

**Transformers for vision-language multimodal learning.** Vision-language multimodal learning has shown remarkable achievement on various tasks, such as classification (Liu et al., 2018; Liang et al., 2022a), retrieval (Fei et al., 2021), few-shot learning (Tsimpoukelli et al., 2021; Alayrac et al., 2022), visual QA (Kim et al., 2016; Liu et al., 2023), and image/video generation (Zhou et al., 2022b; Singer et al., 2022; Lee et al., 2023). Recently, transformer modules have become the new standard for both language and visual-based (Dosovitskiy et al., 2020; Liu et al., 2021) tasks, and the powerful representation of these transformers allows for the modularization of multimodal learning frameworks with existing pre-trained uni-modal models without an expensive re-training phase for multimodal data (Radford et al., 2021; Zhou et al., 2022a).

## 3 BACKGROUND: LAYER-WISE PRUNING FOR VISION-LANGUAGE MODELS

### 3.1 PRELIMINARIES: A FRAMEWORK FOR VISION-LANGUAGE MODELS

Multiple vision-language multimodal models introduce different architectural designs to understand compositional multimodal semantics, which can affect the pruning strategy. In this paper, we basically build our pruning method upon two popular multimodal learning frameworks for vision-language tasks, BLIP (Li et al., 2022) and BLIP-2 (Li et al., 2023c), which are modularized with the pre-trained ViT (Dosovitskiy et al., 2020) and FlanT5 (Chung et al., 2022) architectures to encode visual and language information, respectively. They have achieved remarkable zero-shot performance on various vision-language tasks. Let the visual encoder $f_v(\cdot; \mathbf{W}^v)$ and language model $f_l(\cdot; \mathbf{W}^l)$ in BLIP variants be composed of a set of the weights $\mathbf{W}^v = \{\mathbf{W}_i^v | 1 \leq i \leq M\}$ and $\mathbf{W}^l = \{\mathbf{W}_i^l | 1 \leq i \leq N\}$, respectively, where $M$ and $N$ indicate the number of the corresponding layers. While freezing these pre-trained backbones, BLIPs introduce Query transformer (Q-Former) $f_q(\cdot; \mathbf{W}^q)$, a lightweight module with a set of the weights $\mathbf{W}^q$, that extracts essential information from visual representation and warps it to the input space of the language model (Li et al., 2021). Given the multimodal input pair $(\mathbf{x}^v, \mathbf{x}^l)$, the output of the model is represented as $f_l([\mathbf{o}^v, \mathbf{x}^l]; \mathbf{W}^l)$, where $\mathbf{o}^v$ denotes the aligned visual representation $f_q(f_v(\mathbf{x}^v; \mathbf{W}^v); \mathbf{W}^q)$.

### 3.2 CHALLENGES IN PRUNING VISION-LANGUAGE MODELS

Let us first describe a primary strategy for Hessian-based global pruning, which aims to remove the optimal subset[1] of parameters minimizing the loss change $\delta\mathcal{L} = \mathcal{L}(\mathbf{w} + \delta\mathbf{w}) - \mathcal{L}(\mathbf{w})$. Here, $\mathbf{w}$ denotes the weight vector of the model. When the $i^{th}$ model parameter is removed, implying $\mathbf{e}_i^\intercal \delta\mathbf{w} + \mathbf{w}_i = 0$, where $\mathbf{e}_i^\intercal$ indicates the $i^{th}$ canonical basis vector, its importance is represented as $\mathbf{w}_i^2/(2 \cdot [\mathbf{H}^{-1}]_{ii})$. After that, the model prunes its weights with the lowest importance. However, computing the inverse Hessian $\mathbf{H}^{-1}$ requires significant computational budgets and is often infeasible for large models.

On the other hand, layer-wise one-shot pruning aims to leverage a small amount of calibration data (such as 128 data points) to remove less important weights from the model in a single step

---

[1]The subset size (i.e., sparsity) is often pre-defined or obtained through training.

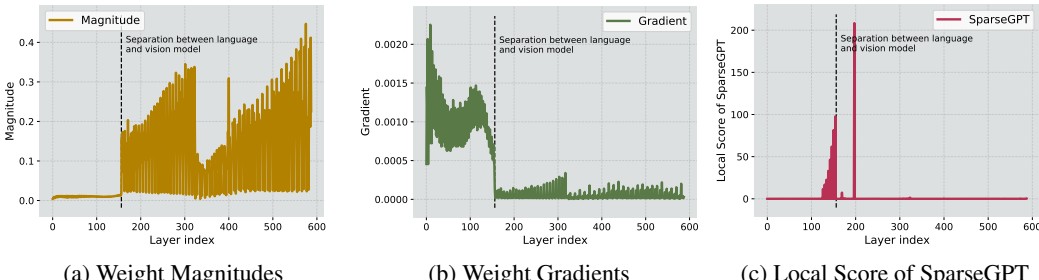

(a) Weight Magnitudes  (b) Weight Gradients  (c) Local Score of SparseGPT

Figure 1: **(a)** and **(b): The imbalance of the magnitude and gradient distributions** between vision and language models. **(c): The skewed distribution of the layer-wise scores** of SparseGPT.

based on their local significance per layer. This approach is significantly efficient in terms of computation/memory in the model pruning phase without expensive iterations of the re-training or Hessian matrix computation for the parameters of the entire model, which is particularly beneficial for large language/multimodal models with billions of parameters. The model compression procedure of recent layer-wise pruning methods (Lazarevich et al., 2021; Yu & Xiang, 2023; Frantar & Alistarh, 2023) can be summarized in the three steps given a target sparsity of $p\%$ and a layer index $i = 1,\ i \in \{1, ..., L\}$: (1) Compute the importance score for each parameter within the layer $i$, (2) Discard $p\%$ of the parameters of layer $i$ based on their importance score and adjust the remaining weights (usually for Hessian-based methods), and (3) calculate the layer's output with only activated (i.e., not pruned) weights, and repeat the process for the next layer ($i \leftarrow i + 1$) until $i = L$.

However, most layer-wise pruning approaches focus on removing unnecessary weights or parameters from single-modal models, such as vision-only (Chen & Zhao, 2018; Lazarevich et al., 2021; Yu & Xiang, 2023) or language-only (Frantar & Alistarh, 2023; Sun et al., 2023a) architectures, which face significant limitations when extended to multimodal (e.g., vision-language) models due to the critical gap in weight/gradient distributions between different modalities, as shown in Figures 1a and 1b. Unlike global pruning methods estimating the importance of each parameter based on the objective loss, layer-wise pruning without the guidance of global weight correlations suffers from obtaining relative significance across weights in different layers/modules due to such severe distributional imbalances in the different modality modules. For example, in Wanda, inputs for different layers significantly vary in their scales (Ba et al., 2016; Ioffe & Szegedy, 2015), and in SparseGPT, the importance of the local loss functions for different layers is also incomparable (Please see the skewed distribution of the local score shown in Figure 1c). This problem makes layer-wise pruning approaches usually set the pruning ratio to be a fixed value for all layers, resulting in a suboptimal compression rate and performance. This is also evident in the results from Singh & Alistarh (2020) that the global WoodFisher pruner outperforms its layer-wise counterpart by a large margin, especially at high sparsity regimes.

## 4 ECoFLaP: Efficient Coarse-to-Fine Layer-Wise Pruning

As discussed above, deploying a layer-wise pruning approach for large vision-language models is challenging due to three primary issues; *distributional/architectural disparity between different modalities*, *significantly large model size*, and *lack of global knowledge*. To overcome such critical challenges, we propose **Efficient Coarse-to-Fine Layer-wise Pruning** (**ECoFLaP**), to efficiently compute an adaptive pruning ratio for each layer with global importance scores (*Coarse*), and then accurately remove the parameters layer by layer with local significance (*Fine*).

### 4.1 THE FRAMEWORK OF COARSE-TO-FINE LAYER-WISE PRUNING

Our focus is to prune the BLIP-like multimodal architectures, as discussed in Section 3.1. Note that we compress the vision and language model in BLIP variants while keeping Q-Former intact because it is sufficiently lightweight, occupying only $\sim 5\%$ of the parameters in the whole framework. Given the multimodal calibration data $\mathcal{D} = \{\mathbf{x}_k^v, \mathbf{x}_k^l\}_{k=1}^K$, a conventional layer-wise pruning method aims to

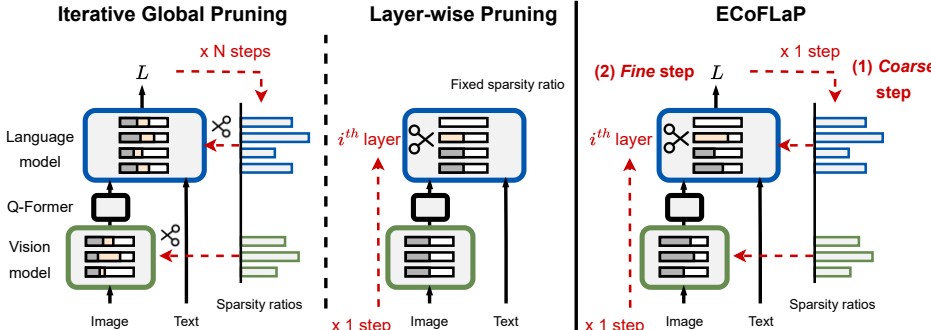

Figure 2: **Illustration of our ECoFLaP** compared to *global* and *layer-wise pruning*. The boxes with a **blue**, **green**, and **black** border denote the language, vision, and Q-Former modules, respectively. The dotted red arrows show the working flow of the algorithm. The **beige** color indicates the pruning of the current step (layer) is conditioned on the pruning decisions made in the preceding steps (layers), which are marked in **gray**. ECoFLaP first performs the efficient *coarse* step to obtain the pruning ratio for each layer by leveraging the zeroth-order gradient, and then removes the uncritical weights in a layer-wise fashion in the *fine* step.

find the sparse weight $\widehat{\mathbf{W}}_i$ at layer $i$ via the corresponding local objective $\mathcal{L}_i$:

$$\widehat{\mathbf{W}}_i = \arg\max \mathcal{S}(\mathbf{W}_i|\widehat{\mathbf{W}}_{i-1}, \mathcal{D}, \mathcal{L}_i), \quad \text{s.t.,} \quad \frac{|\widehat{\mathbf{W}}_i|}{|\mathbf{W}_i|} = p_i, \tag{1}$$

where $\mathcal{S}(\cdot)$ is the score function to compute the importance of the weight, $\widehat{\mathbf{W}}_i$ is denoted as the pruned weight from $\mathbf{W}_i$, and $p_i\%$ the desired sparsity at layer $i$. Basically, $p_i$ is the same for all layers in layer-wise pruning methods.

In our proposed Efficient Coarse-to-Fine Layer-wise Pruning (ECoFLaP) framework, we estimate the optimal sparsity for each layer in the vision-language model by computing the global importance of model weights. One straightforward direction to obtain the layer sparsities is to perform global pruning on the model for the target global sparsity $p$ by leveraging the global loss objective $\mathcal{L}$,

$$\widehat{\mathbf{W}} = \arg\max \mathcal{S}(\mathbf{W}|\mathcal{D}, \mathcal{L}), \quad \text{s.t.,} \quad \frac{|\widehat{\mathbf{W}}|}{|\mathbf{W}|} = p. \tag{2}$$

Then, we can estimate the layer sparsity via $p_i = |\widehat{\mathbf{W}}_i|/|\mathbf{W}_i|$. However, this approach requires an expensive operation to obtain gradients for all weights in the large model and to extract the pruned weights via `argmax`, consuming enormous memory and computational resources. Therefore, we introduce an alternative numerical approach, which computes the keep ratio (1 - sparsity ratio) based on importance scores linearly, to avoid expensive operations that can estimate the global importance score efficiently. First, we obtain the importance score for each weight via the global objective function, namely, $\mathbf{s} = \mathcal{S}(\mathbf{W}|\mathcal{D}, \mathcal{L})$, and then convert the scores to sparsity by three steps: (1) Find the total parameters that need to be selected based on $p$, (2) Normalize the scores, (3) Obtain the sparsity for each layer based on the number of parameters to be picked and the parameters of this layer. The example to determine the sparsity ratio for the $i^{th}$ layer of the model is shown below,

$$\text{normalize}(\mathbf{s}_i, \mathbf{s}) = \frac{\mathbf{s}_i}{\sum \mathbf{s}}, \tag{3}$$

$$p_i = 1 - \frac{(\text{normalize}(\mathbf{s}_i, \mathbf{s}) \cdot N_{\text{select}})}{|\mathbf{W}_i|}, \quad \text{where } N_{\text{select}} = (1 - p) \cdot |\mathbf{W}|. \tag{4}$$

We then inject the derived sparsity ratios back to Equation 1 to prune the model layer-wisely. We also introduce a hyperparameter $p_{max}$ to control the maximum sparsity for each layer to avoid the layer collapse, which is discovered by previous literature (Tanaka et al., 2020) that the global pruning approach may remove all the parameters for one layer. To incorporate this hyperparameter in the Equation 3, we simply pre-pick the parameters for each layer to satisfy the max sparsity condition, subtracting the $N_{select}$ with the number of pre-picked parameters, and start the algorithm. In our experiments, we find using the same sparsity ratio for the layers in one block (such as the feed-forward layer and attention layer in a transformer block) makes the results more robust, so we also extend the

algorithm to determine group sparsity ratios by via group scores (aggregating all the layer scores in a block). For the layer-wise pruning step, we utilize Wanda, which combines the input activation of the layer with the weight magnitude as the local importance score, to remove uncritical parameters based on the obtained layer sparsity ratios.

## 4.2 ZEROTH-ORDER GLOBAL INFORMATION

In the previous section, we introduce our coarse-to-fine pruning framework to determine the pruning ratios for layer-wise pruning. Note that our goal is to prune models with billions of parameters, so we again focus on the one-shot setting in this step without any iterative refinement. To this end, our first attempt is to use the first-order backward gradient with weight magnitude (Le-Cun et al., 1989), $|\mathbf{W}| \cdot |\nabla_{\mathbf{W}}\mathcal{L}(\mathbf{W}, \mathcal{D})|$, as the global importance score, and the performance improvement demonstrates the effectiveness of our framework (Please see Table 1 and Figure 4). However, even though the first-order backward gradient is still tractable for most models. like less than 5B parameters, 7B LLaMA (Touvron et al., 2023) requires more than **40GB** even with float16 weight type and batch size being one.

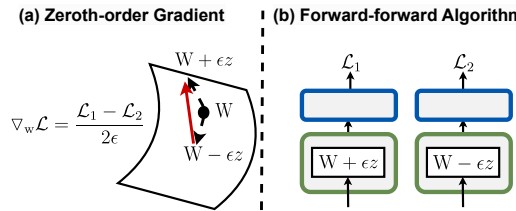

Figure 3: Illustration of (a) the zeroth-order gradient in weight space, and (b) the forward-forward algorithm.

To further reduce the cost of computing gradients, we instead compute the zeroth-order approximated gradient by replacing the backpropagation with the forward-forward algorithm (Hinton, 2022; Malladi et al., 2023). We take the $i^{th}$ layer of a model as an example, the zeroth-order gradient of $\mathbf{W}_i$ can be computed by perturbing the weight with Gaussian noises twice, that is,

$$\|\nabla_{\mathbf{W}_i}\mathcal{L}(\mathbf{W}_i, \mathcal{D})\|_2 = \mathbb{E}_{d\sim\mathcal{D}}[\mathbb{E}_{z\sim N(0,1)}[|\frac{\mathcal{L}(\mathbf{W}_i + \epsilon z, d) - \mathcal{L}(\mathbf{W}_i - \epsilon z, d)}{2\epsilon}|]] \quad (5)$$

We hence use $\|\nabla_{\mathbf{W}_i}\mathcal{L}(\mathbf{W}_i, \mathcal{D})\|_2$ as the importance score of $\mathbf{W}_i$. Note that we do not multiply it with the weight magnitude as solely using the approximated gradients works better empirically. The computation of Equation 5 is extremely memory-efficient and the exact process for the $i^{th}$ layer is shown in the following: (1) First, we choose a random seed and then sample a noise $z$ (now we use 2x of memory of "this layer"), (2) we modify the model's weight with $\mathbf{W} + \epsilon z$ and compute the loss change $\mathcal{L}(\mathbf{W} + \epsilon z, d)$, (3) we then sample the same noise $z$ as the first step (by setting the same random seed), and we modify the model's weight with $\mathbf{W} - \epsilon z$, which can be done by subtracting $2\epsilon z$ from the previous weight, and compute $\mathcal{L}(\mathbf{W} - \epsilon z, d)$, (4) obtain the gradient norm by $(\mathcal{L}(\mathbf{W} + \epsilon z) - \mathcal{L}(\mathbf{W} - \epsilon z))/(2\epsilon)$ (all terms are scalars). From the above process, for an N-layer network, we only need (N+1) times of one-layer memory, which is significantly memory-efficient than storing the whole gradients that cause 2N times of one-layer memory. This explains why our approach uses much less memory than first-order based methods, which also need to cache input activations. One may suspect we need to sample many noises to capture the loss of the landscape and to estimate the gradient accurately. However, our experiments and the previous studies (Malladi et al., 2023) show that one noise is sufficient. We describe our approach in detail in Algorithm 1.

## 5 EXPERIMENTAL SETUP

We report evaluation metrics and more experimental details in Appendix C.

**Architectures.** We use multiple uni- and multi-modal architectures for experiments: the encoder-decoder vision of BLIP-2 (Li et al., 2023c), composed of pre-trained EVA-ViT (ViT-g/14 from EVA-CLIP) (Sun et al., 2023b) and FlanT5 (Chung et al., 2022), is used for most experiments and ablation studies. And, we also extend our approach to BLIP (Li et al., 2022) with ViT (Dosovitskiy et al., 2020) and BERT (Devlin et al., 2019) backbones. In addition, we evaluate our approach solely on unimodal vision and NLP tasks with EVA-ViT, FlanT5, and LLaMA backbones.

**Evaluation Datasets.** We evaluate the zero-shot ability of BLIP-2 on various datasets after pruning, such as VQAv2, OK-VQA, and GQA for visual question answering, NoCaps for image captioning, and Flickr30k for image-text retrieval. For BLIP, we evaluate the performance change of the BLIP

Table 1: Comparison of existing pruning approaches with ECoFLaP on the zero-shot performance with BLIP-2 at 0.5 sparsity. We report accuracy for visual question answering, CIDEr and SPICE for image captioning, and TR@1 (text recall) and IR@1 (image recall) for image retrieval. We also compute the macro average score over tasks and the GPU memory usage for each method.

| Method | Sparsity | Method's Mem. Usage (GB) | Visual Question Answering | | | Image Captioning | | Image retrieval | | Macro Avg. |
| | | | VQAv2 | OK-VQA | GQA | NoCaps | | Flickr30k | | |
| | | | | Accuracy | | CIDEr | SPICE | TR@1 | IR@1 | |
| Full Model | 0% | - | 63.1 | 41.1 | 44.1 | 121.7 | 15.8 | 97.6 | 89.7 | 62.1 |
| Global Magnitude Pruning | 50% | 8.46 | 0.0 | 0.0 | 0.0 | 0.0 | 0.0 | 0.2 | 0.1 | 0.0 |
| Gradient-based Pruning | 50% | 22.4 | 56.9 | 35.3 | 41.7 | 96.8 | 13.5 | 93.0 | 82.4 | 55.4 |
| SparseGPT | 50% | 9.04 | 57.4 | 36.1 | 40.8 | 108.1 | 14.1 | **96.4** | **86.3** | 57.4 |
| Wanda | 50% | 8.87 | 57.9 | 35.3 | 42.2 | 106.9 | 14.1 | 95.1 | 84.6 | 57.2 |
| ECoFLaP | | | | | | | | | | |
| + First-order Gradient | 50% | 22.4 | **59.4** | **36.3** | 42.3 | 109.9 | 14.2 | 95.9 | 86.2 | **58.2** |
| + Zeroth-order Gradient | 50% | 8.93 | 58.4 | 35.6 | **42.9** | **110.2** | **14.3** | 95.5 | 85.8 | 58.0 |

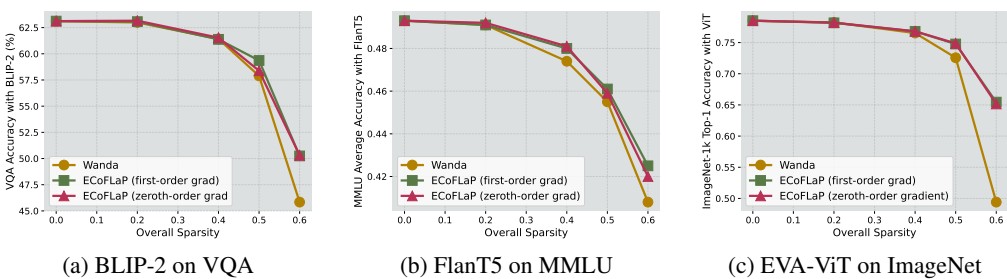

(a) BLIP-2 on VQA  (b) FlanT5 on MMLU  (c) EVA-ViT on ImageNet

Figure 4: The accuracy and model sparsity trade-off for Wanda, ECoFLaP with first-order and zeroth-order gradient on both BLIP-2 (multimodal), FlanT5 (unimodal), and EVA-ViT (unimodal).

fine-tuned on NLVR$^2$ and COCO captions. For unimodal models, we evaluate FlanT5 on MMLU, evaluate LLaMA 7B with WikiText, and evaluate EVA-ViT with ImageNet-1k.

**Baselines.** We compare ECoFLaP to several global pruning and layer-wise pruning approaches. For **Global Magnitude Pruning**, we perform global pruning based on the magnitude of the weight. **Gradient-based Pruning** is an iterative global pruning approach, where we use the multiplication of the first-order gradient and the weight magnitude as the importance score and prune the model to target sparsity in 3 iterations. **SparseGPT** is a layer-wise Hessian-based method. **Wanda** is also a layer-wise approach utilizing the multiplication of the weight magnitude and the norm of input activation as the local importance score. The comparison of the above pruning methods and their importance metric can be found in Table 8. We also compare our method with **UPop**, which prunes and re-trains the vision-language model simultaneously in a unified progressive pruning manner.

# 6 RESULTS

## 6.1 MAIN EXPERIMENTAL RESULTS

We demonstrate the zero-shot performance on various datasets using BLIP-2 pruned by our ECoFLaP and baselines at a 0.5 sparsity ratio in Table 1. And we compare our approach with UPop using BLIP backbone on NLVR$^2$ and COCO captions under the fine-tuning and non-fine-tuning settings in Table 3. We also provide the comparison of our approach and Wanda at various sparsities on both multi- and uni-modal models in Figure 4. We summarize our observations as follows:

**ECoFLaP consistently outperforms Global Magnitude Pruning (GMP) and Gradient-based Pruning** as shown in Table 1. GMP fails to generate meaningful results at 0.5 sparsity. In particular, we find that GMP suffers from pruning the vision module, EVA-ViT, so the degraded visual representation harms the multimodal model performance. Gradient-based Pruning performs significantly better by selecting crucial weights based on gradient information of the whole model, but it requires a larger memory than GMP due to the full pass to compute the gradients. On the other hand, compared to Gradient-based Pruning, our ECoFLaP with zeroth-order gradients uses only $40\%$ memory

Table 2: Zero-shot evaluation on 11 classification tasks with CLIP using $0.4$ of sparsity ratio. We additionally report results using local scores to compute layer sparsity ratios. *FGVC, OFlowers, OPets, and SCars stand for FGVCAircraft, OxfordFlowers, OxfordPets, and StanfordCars, respectively.

| Methods | Caltech101 | DTD | EuroSAT | FGVC* | Food101 | ImageNet | OFlowers* | OPets* | SCars* | SUN397 | UCF101 | Avg |
|---|---|---|---|---|---|---|---|---|---|---|---|---|
| Full Model | 92.9 | 44.5 | 47.8 | 24.8 | 86.1 | 66.7 | 71.3 | 89.1 | 65.3 | 62.6 | 66.8 | 65.3 |
| Wanda | 85.1 | 31.2 | 33.5 | 9.0 | 66.0 | 45.6 | 37.2 | 71.3 | 35.0 | 50.8 | 54.7 | 47.2 |
| SparseGPT | 90.5 | 41.5 | 47.4 | 19.2 | 79.4 | 50.8 | 59.0 | 86.4 | 53.4 | 55.3 | 63.2 | 58.7 |
| Sparsities from Local Scores | | | | | | | | | | | | |
| w/ Wanda | 84.0 | 30.1 | 31.5 | 6.6 | 62.9 | 44.5 | 34.2 | 68.1 | 34.1 | 49.5 | 52.9 | 45.3 |
| w/ SparseGPT | 88.6 | 37.7 | 43.1 | 17.7 | 76.6 | 42.1 | 50.3 | 83.2 | 46.1 | 48.0 | 60.4 | 54.0 |
| ECoFLaP | | | | | | | | | | | | |
| w/ Wanda | 87.5 | 37.9 | 43.8 | 16.0 | 72.8 | 53.5 | 57.0 | 80.3 | 46.9 | 58.2 | 62.4 | 56.0 |
| w/ SparseGPT | 90.0 | 42.4 | 45.4 | 22.5 | 81.2 | 56.5 | 64.1 | 88.3 | 56.8 | 60.2 | 63.4 | 61.0 |

during pruning while improving average accuracy by $2.12\%$, $9.88\%$, and $3.41\%$, on visual question answering, image captioning, and image retrieval tasks, respectively.

**ECoFLaP outperforms SoTA layer-wise pruning baselines** since our ECoFLaP is able to exploit the global information. Our approach with zeroth-order gradient consistently achieves better performance against Wanda on all tasks and metrics by $1.1\%$, $2.3\%$, and $1.0\%$, on visual question answering, image captioning, and image retrieval tasks, respectively, showing the effectiveness of our coarse-to-fine framework. Furthermore, we highlight that the gap between Wanda and ECoFLaP becomes larger when the sparsity increases to 0.6, where ECoFLaP outperforms Wanda by 9.6% on VQA (shown in Figure 4a). Our approach also surpasses SparseGPT by 1.4% of relative improvement on the average score. Note that our zeroth-order ECoFLaP uses on-par memory budgets compared to layer-wise pruning methods, Wanda and SparseGPT, by avoiding expensive backpropagation computation. Additional results on the combination of ECoFLaP and SparseGPT are in Table 9.

**ECoFLaP surpasses baselines on pruning CLIP models by a significant margin**. Beyond pruning modularized VL models, ECoFLaP can also work well with joint VL models like CLIP. In Table 2, we evaluate Wanda, SparseGPT, and our approach on CLIP by measuring the zero-shot accuracy on 11 classification tasks before and after pruning over the sparsity $0.4$, and our ECoFLaP excels against baselines while effectively mitigating the degeneration of zero-shot accuracy through the model pruning. In addition, we assess the impact of using global scores for computing layer sparsity ratios by comparing it with the approach using local scores derived from layer-wise pruning methods (namely, Wanda and SparseGPT). **The result shows that the strategy of using local scores leads to a decrease in performance compared to leveraging a uniform sparsity ratio**. This observation emphasizes the ineffectiveness of employing local scores for assessing layer importance (Please see our motivation in Figure 1).

**ECoFLaP outperforms the multimodal model pruning baseline, UPop, in both non-fine-tuning and fine-tuning cases** as shown in Table 3. Note that UPop requires re-training the backbone model in the pruning phase, which is significantly expensive in memory and computation, especially for large models. On the contrary, ECoFLaP finds an important subset of weights in the model in a single shot and then fine-tunes the pruned model. We also emphasize that our approach generalizes to different architectures and datasets, where BLIP uses a BERT-encoder for $NLVR^2$ (classification) and a BERT-decoder for COCO (image captioning).

Table 3: Performance comparison on BLIP at $0.5$ sparsity on $NLVR^2$ and COCO captions.

| Method | $NLVR^2$ | | COCO cap. | |
|---|---|---|---|---|
| | val | test | CIDEr | SPICE |
| Full Model | 82.3 | 83.6 | 133.3 | 23.8 |
| *w/o fine-tuning* | | | | |
| UPop | 76.9 | 77.8 | - | - |
| Wanda | 78.3 | 78.1 | 97.1 | 18.4 |
| ECoFLaP | **79.1** | **79.2** | **111.0** | **20.3** |
| *w/ fine-tuning* | | | | |
| UPop | 80.3 | 81.1 | 128.9 | 23.3 |
| ECoFLaP | **81.8** | **82.5** | **132.3** | **23.8** |

**ECoFLaP also generalizes well to unimodal tasks/architectures.** In Figure 4b and Figure 4c, we visualize the accuracy of MMLU with FlanT5 and ImageNet-1k with EVA-ViT after pruning with various sparsity ratios (from $0.1$ to $0.6$). Since Wanda applies the same pruning ratio to all layers, performance degrades with increasing sparsity ($\sim 40\% \uparrow$), which is catastrophic for some layers containing much more informative weights than others. Alternatively, the performance degradation caused by pruning weights can be significantly mitigated by proposing adaptive layer-wise pruning with dynamic sparsity for each layer based on estimated global weight importance. Furthermore, as shown in Table 7, our approach shows the ability to generalize to larger LLM, where we attain 10.6% relative improvement over Wanda when evaluated on WikiText using perplexity with LLaMA 7B.

Table 4: The ablation on the maximum sparsity.

| BLIP-2 | Max Sparsity | | | |
|---|---|---|---|---|
| $(p = 0.5)$ | 0.5 | 0.6 | 0.8 | 1.0 |
| VQA | 57.9 | **58.4** | 57.1 | 51.6 |

Table 5: The ablation on number of samples.

| BLIP-2 | Number of Samples | | | |
|---|---|---|---|---|
| $(p = 0.5)$ | 16 | 32 | 64 | 128 |
| VQA | 58.2 | **58.4** | 58.3 | **58.4** |

Table 6: The ablation on number of noises (total samples = 32).

| BLIP-2 | Number of noises | | | |
|---|---|---|---|---|
| $(p = 0.5)$ | 1 | 4 | 8 | 32 |
| VQA | **58.4** | 58.1 | 58.0 | 58.1 |

## 6.2 ABLATIONS AND VISUALIZATION

We provide ablation studies of our zeroth-order ECoFLaP on maximum sparsity for each layer $p_M$, the number of data $|\mathcal{D}|$, and the number of perturbed noises per sample. We report the zero-shot VQA accuracy obtained from the BLIP-T5 with 0.5 total sparsity. Note that we use the numerical method to obtain the sparsity ratios for experiments (Please see Equation 3).

**Maximum Sparsity.** First, we observe that the maximum sparsity is critical as it affects the results considerably, and the optimal value happens at $0.6$, which is a conservative value given the target ratio is $0.5$. Similar to the findings discovered by Tanaka et al. (2020), the algorithm might suffer layer collapse when the maximum sparsity is not set ($p_M = 1$). We find it simple but effective to introduce this hyperparameter as we would like to avoid using the iterative solution proposed by Tanaka et al. (2020) because we are targeting the one-shot pruning. Based on the result shown in Table 4, we simply set $p_M = p + 0.1$ throughout our experiments.

**Number of Data and Noises.** In the next ablation experiment, we focus on understanding how many samples and noises we need to estimate the gradient with zeroth-order optimization. We first ablate the effect of the number of data by fixing the number of noises per sample to be one, and the result in Table 5 shows that our approach is robust to the number of samples across 16 to 128, and we thus using 32 samples for better efficiency. Next, we explore if having a more accurate gradient estimation by sampling more noises can improve the accuracy. In this ablation, we make sure to forward through the model the same number of times when using different numbers of noises, meaning that we cut the number of data to $1/n$ when we increase the number of noises to $n$. The result in Table 6 demonstrates that using more data samples is more effective compared to sampling more noises when the budget is fixed.

**Visualization and Analysis.** We also provide the visualization of the sparsity ratios obtained by ECoFLaP (Figure 6) and the loss landscape of BLIP-2 (Figure 5) to justify the use of zeroth-order optimization. In Figure 6, we observe that our ECoFLaP variants allocate sparsity ratios in a similar distribution, in favor of lower sparsity ratios for the visual model and higher sparsity ratios for the language model, and this alignment provides evidence that the zeroth-order gradient obtained by the forward-forward algorithm can be the accurate but cheaper alternative to the first-order gradient in importance estimation. The landscape shown in Figure 5 presents that BLIP-2 is located at a smooth basin with a single local minimum. This loss landscape justifies the success of utilizing zeroth-order optimization as we avoid sampling weights located in other basins that may cause an inaccurate gradient estimation. We also present a qualitative analysis of the pruned model in Appendix E.

## 7 CONCLUSION

This paper has investigated the challenges of pruning large-scale vision-language models due to the dilemma between immense memory/computational overhead from global pruning and the suboptimal model performance of layer-wise pruning. Existing global pruning methods require an expensive backpropagation process to compute the inverse Hessian to obtain global weight importance, which makes these approaches infeasible for current large LLMs and large LVLMs. On the other hand, layer-wise pruning approaches are highly efficient by pruning weights based on layer-wise local information, but often suffer from a lack of global importance, resulting in a significant performance degeneration. To address the limitations of these pruning approaches while enjoying their merits, we propose ECoFLaP that adaptively finds the optimal sparsity for each layer by efficiently estimating the global weight importance, and then accurately prunes the vision-language model in a layer-wise manner dependent on the obtained pruning ratios. We utilize the zeroth-order optimization to obtain the gradients with forward pass only and achieve superior performance over multiple uni- and multi-modal tasks against recent layer-wise pruning methods by a significant margin, while consuming only around 40% of GPU memory compared to the iterative global pruning approach. We hope our new efficient pruning can help deploy quality yet compact vision-language models.

# 8 REPRODUCIBILITY STATEMENT

Our codes are based on the publicly available LAVIS (Li et al., 2023b), Wanda (Sun et al., 2023a), MMLU (Hendrycks et al., 2021), UPop (Shi et al., 2023), and CoOp (Zhou et al., 2021). The experimental setup and details can be found in Section 5 and Appendix C, respectively. We also have made our code publicly available.

ACKNOWLEDGMENTS

We thank Prateek Yadav for helpful discussions. This work was supported by ARO Award W911NF2110220, ONR Grant N00014-23-1-2356, and NSF-AI Engage Institute DRL-211263. The views, opinions, and/or findings contained in this article are those of the authors and not of the funding agency.

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

---

**Algorithm 1:** Efficient Coarse-to-Fine Layer-wise Pruning

---

**Input:** weights of the vision model $\mathbf{W}^v$, weights of the language model $\mathbf{W}^l$, calibration dataset $\mathcal{D}$, target sparsity $p$, maximum sparsity per layer $p_{max}$

**Output:** The pruned weights $\widehat{\mathbf{W}}^v$ and $\widehat{\mathbf{W}}^l$

    // *Coarse* step: determine the sparsity ratios via global information

1   $\mathbf{s} \leftarrow \mathcal{S}(\mathbf{W}^v, \mathbf{W}^l, \mathcal{D}, \mathcal{L})$ ;                             // get the global importance score

2   $\mathbf{p} \leftarrow \text{getSparityFromScores}(\mathbf{s}, \mathbf{W}^v, \mathbf{W}^l, p, p_{max})$ ;     // obtain the sparsity via Equation 3

    // *Fine* step: perform layer-wise pruning

3   $\widehat{\mathbf{W}}^v, \widehat{\mathbf{W}}^l = \{\}, \{\}$

    // We initialize the $0^{th}$ weight to be the identity matrix, depicting the fact that the input of the first layer is the raw input.

4   $\mathbf{W}_0^v \leftarrow I$

5   **for** $i = 1$ *to* $M$ **do**

6      $\widehat{\mathbf{W}}_i^v = \arg\max \mathcal{S}(\mathbf{W}_i^v | \widehat{\mathbf{W}}_{i-1}^v, \mathcal{D}, \mathcal{L}_i^v, \mathbf{p}_i^v)$ ;     // prune vision weights layer by layer

7      $\widehat{\mathbf{W}}^v = \widehat{\mathbf{W}}^v + \{\widehat{\mathbf{W}}_i^v\}$

8   $\mathbf{W}_0^l \leftarrow \mathbf{W}_M^v$

9   **for** $i = 1$ *to* $L$ **do**

10      $\widehat{\mathbf{W}}_i^l = \arg\max \mathcal{S}(\mathbf{W}_i^l | \widehat{\mathbf{W}}_{i-1}^l, \mathcal{D}, \mathcal{L}_i^l, \mathbf{p}_i^l)$ ;     // prune language weights layer by layer

11      $\widehat{\mathbf{W}}^l = \widehat{\mathbf{W}}^v + \{\widehat{\mathbf{W}}_i^l\}$

12   **return** $\widehat{\mathbf{W}}^v, \widehat{\mathbf{W}}^l$

---

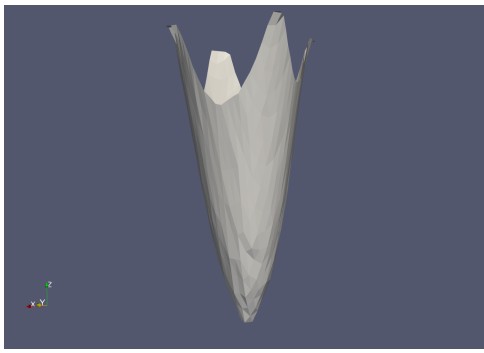

Figure 5: The loss landscape of the BLIP-2 model.

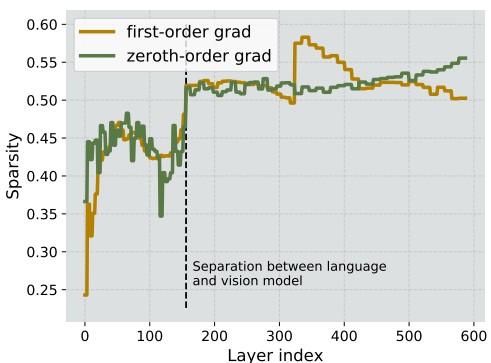

Figure 6: Sparsity ratios obtained by ECoFLaP.

## A  THE LOSS LANDSCAPE

We present a visualization of the loss landscape (Li et al., 2018) for BLIP-2 in Figure 5, where we achieve this by perturbing the pre-trained BLIP-2 model 2500 times to approximate the parameter space and using 256 samples from CC3M to compute the loss. Our visualization reveals a distinctive corn-shaped loss landscape, which is characterized by a single basin centered around the pre-trained weight of the BLIP-2. Moreover, the landscape is surprisingly smooth with a linear slope around the pre-trained weight. This smooth and corn-shaped loss landscape explains why we can estimate the gradient with zeroth-order optimization this successfully, as we avoid sampling weights situated in other basins that could lead to inaccurate gradient estimation.

## B  RESULTS ON LLAMA

We further assess our approach's performance using perplexity on the WikiText dataset with LLaMA, a prominent open-source LLM. The results on the 7B model presented in Table 7 demonstrate that ECoFLaP achieves a noteworthy 10.6% relative improvement compared to Wanda when operating

Table 7: Evaluation of Wanda and ECoFLaP on LLaMA 7B at 0.6 sparsity by perplexity on WikiText.

| LLaMA @ 0.6 sparsity | Perplexity |
|---|---|
| Full Model | 7.26 |
| Wanda | 10.68 |
| ECoFLaP | |
| + first-order gradient | 10.16 |
| + zeroth-order gradient | 9.83 |

at 0.6 sparsity. Notably, in this experiment, we find that employing a larger $\epsilon$ value in Equation 5 substantially enhances performance. This observation suggests that the zeroth-order estimation can benefit from exploring regions further from the target weight, a capability not inherent in gradients obtained through backpropagation.

## C   EXPERIMENTAL DETAILS

**Evaluation Datasets and Metrics.** We evaluate the zero-shot ability of BLIP-2 on various datasets after pruning. We use VQAv2 (Goyal et al., 2016), OK-VQA (Marino et al., 2019), and GQA (Hudson & Manning, 2019) for visual question answering, NoCaps (Agrawal et al., 2019) for image captioning, and Flickr30k (Plummer et al., 2015) for image-text retrieval. For BLIP, we evaluate the performance change of the BLIP, fine-tuned on NLVR$^2$ (Suhr et al., 2019) and COCO captions (Chen et al., 2015). For unimodal models, we evaluate FlanT5 on 57 tasks in MMLU (Hendrycks et al., 2021), evaluate LLaMA 7B with WikiText (Merity et al., 2016), and evaluate EVA-ViT on image classification task with ImageNet-1k (Deng et al., 2009). We report accuracy for question-answering datasets, NLVR$^2$, ImageNet-1k, and tasks in MMLU. Next, we use CIDEr and SPICE to evaluate image captioning tasks and use TR@1 (top-1 text recall) and IR@1 (top-1 image recall) for image retrieval tasks. Lastly, we report the model's perplexity of WikiText in evaluating the LLaMA.

**Calibration Datasets.** Our approach utilizes a small subset (128 samples) of a dataset as the calibration data from CC3M (Sharma et al., 2018), ImageNet (Deng et al., 2009) and C4 (Raffel et al., 2019) for calibrating BLIP-2, unimodal EVA-ViT, and unimodal FlanT5 (and LLaMA). We use NLVR$^2$ and COCO captions as the calibration data for fine-tuning the BLIP backbone.

**Dataset splits.** For VQAv2, OK-VQA, and GQA, we use val, test, and test-dev split, respectively. We use the validation set for NoCaps and the test set for Flickr30k. In our BLIP experiments, we report results on both val and test set for NLVR$^2$, while we use the Karpathy test split (Karpathy & Fei-Fei, 2014) for COCO captions. In our unimodal experiments, we use the publicly available test set for MMLU, and we use the validation set for ImageNet-1k. For WikiText, we report the perplexity on the validation set.

**Datasets and Metrics for CLIP experiments.** We use a diverse set of 11 image classification datasets to evaluate the model's ability to recognize generic and specific objects: Caltech101 (Fei-Fei et al., 2004), DTD (Cimpoi et al., 2013), EuroSAT (Helber et al., 2017), FGVCAircraft (Maji et al., 2013), Food101 (Bossard et al., 2014), ImageNet (Deng et al., 2009), OxfordFlowers (Nilsback & Zisserman, 2008), OxfordPets (Parkhi et al., 2012), StanfordCars (Krause et al., 2013), SUN397 (Xiao et al., 2010), UCF101 (Soomro et al., 2012). We report accuracy for each dataset in Table 2.

**Seed.** The default seed option for each codebase is used for evaluation: seed 42 for all the experiments for BLIP-2, FlanT5, and EVA-ViT, seed 0 for the LLaMA experiment, and seed 1 for the CLIP experiment.

**Hyperparameters.** For the coarse step, we set $p_M = p + 0.1$, $|\mathcal{D}| = 32$, and the number of noises to be 1 for ECoFLaP with zeroth-order gradient. $\epsilon$ is set to $1e^{-3}$ except for LLaMA, where we find $1e^{-1}$ works better, but we almost did not tune this hyperparameter. For ECoFLaP with first-order gradient, we use $|\mathcal{D}| = 128$.

**Resources.** All the experiments are done with one 40GB A100 or one 48GB A6000, except for ECoFLaP with first-order gradient on LLaMA we use 2x 48GB A6000.

Table 8: The comparison of different methods in terms of their category and used importance measure.

| Methods | Global or Layer-wise | Importance Measure |
|---|---|---|
| Global Magnitude Pruning | Global | $\|\mathbf{W}\|_{ij}$ |
| Gradient-based Pruning | Global | $\|\mathbf{W}\|_{ij} \cdot \|\nabla_{\mathbf{W}_i}\mathcal{L}\|_{ij}$ |
| SparseGPT | Layer-wise | $[\|\mathbf{W}\|^2 / \text{diag}(\mathbf{XX}^T + \lambda\mathbf{I})^{-1}]_{ij}$ |
| Wanda | Layer-wise | $\|\mathbf{W}_{ij}\| \cdot \|\mathbf{X}_j\|^2$ |
| ECoFLaP | Mixed | First Stage: $\|\nabla_{\mathbf{W}_i}\mathcal{L}\|_{ij}$ |
| w/ Wanda and zeroth-order | | Second Stage: $\|\mathbf{W}_{ij}\| \cdot \|\mathbf{X}_j\|^2$ |

Table 9: Comparison of SparseGPT and ECoFLaP (with SparseGPT) on the zero-shot performance with BLIP-2 at 0.6 sparsity. We report accuracy for visual question answering, CIDEr and SPICE for image captioning, and TR@1 (text recall) and IR@1 (image recall) for image retrieval. We also compute the macro average score over tasks and the GPU memory usage for each method.

| Method | Sparsity | Method's Mem. Usage (GB) | Visual Question Answering | | | Image Captioning | | Image retrieval | | Macro Avg. |
|---|---|---|---|---|---|---|---|---|---|---|
| | | | VQAv2 | OK-VQA Accuracy | GQA | NoCaps CIDEr | SPICE | Flickr30k TR@1 | IR@1 | |
| SparseGPT | 60% | 9.04 | 48.7 | **27.9** | 35.1 | 101.0 | 13.4 | 95.2 | 85.0 | 51.8 |
| ECoFLaP w/ SparseGPT | 60% | 9.05 | **50.4** | **27.9** | **36.3** | **105.3** | **14.0** | **95.3** | **86.1** | **53.1** |

## D  ECoFLaP ON SPARSEGPT

As the key contribution of our method for efficient estimation of layer-adaptive pruning rate for layer-wise pruning is orthogonal to SparseGPT, ECoFLaP can also be combined with SparseGPT. As shown in Table 9, EcoFLaP (w/ SparseGPT) consistently achieves superior performance over multiple VL tasks.

Additionally, the performance improvement seems relatively smaller compared to ECoFLaP with Wanda (i.e., original EcoFLaP). We hypothesize that this is because the reweighting mechanism in SparseGPT is somewhat correlated to our dynamic sparsity ratios, where both of them are important to preserve the performance in high sparsity ratios.

## E  QUALITATIVE COMPARISON OF THE MODEL BEFORE AND AFTER PRUNING

We conducted an analysis comparing the predictions of the Full model (before pruning) and the model pruned by ECoFlaP (with 0.5 sparsity) on the GQA datasets, and presented three illustrative examples in Table 10. We found one interesting observation: out of 12578 questions, the Full model correctly answered 1219 questions that our model did not, yet our model also accurately responded to 1072 questions where the Full model failed to respond (there are 4322 questions that both models answer correctly). This indicates that pruning does not always lead to performance degradation; in some instances, it may even enhance model robustness for certain questions. During our review of incorrect predictions by either our model or the Full model, we noticed that most were conceptually similar to the ground truth. This suggests that the pruned model still retains most of the capability. Furthermore, in some cases, such as with the 'horses running' example, both models provided answers that might be considered correct by many humans, even though they differ from the ground truth.

Table 10: Qualitative examples to compare the predictions generated by the Full Model and the ECoFLaP-pruned model. We pick three example image-question pairs from GQA datasets. We observe the model after pruning does not always perform worse than the Full Model.

| | | | |
|---|---|---|---|
| **Images** |  |  |  |
| **Question ID** | 202174017 | 20984557 | 20602949 |
| **Questions** | What shape is the microwave the stove is below? | Which kind of vehicle is parked on the street? | What is the horse running across? |
| **Ground Truth** | rectangular | car | ground |
| **Full Model's Prediction** | square | car | hill |
| **ECoFLaP-pruned Model's Prediction** | rectangular | bicycle | rocks |

