# OpenReview forum: "ECoFLaP: Efficient Coarse-to-Fine Layer-Wise Pruning for Vision-Language Models"
_ICLR.cc/2024/Conference — ICLR 2024 poster_

### Official Review · Reviewer_q2nJ · 2023-10-30

**Soundness:** 3 good
**Presentation:** 3 good
**Contribution:** 2 fair
**Rating:** 5
**Confidence:** 4

**Summary:**

This article primarily addresses the task of model compression for Large Vision-
Language Models (LVLMs). Traditional iterative global pruning methods involve
computationally expensive operations, while layer-wise pruning approaches lack a
global perspective, potentially resulting in suboptimal performance after pruning. This
paper proposes a two-stage coarse-to-fine weight pruning approach for LVLMs, which
utilizes the global importance score for unstructured pruning.

**Strengths:**

1. This paper introduces a layer-wise pruning method for LVLMs that utilizes
global importance scores. The global importance score is obtained by
approximating the first-order gradients of the model parameters. By
considering the global perspective, this approach aims to effectively identify
and prune less important weights in each layer of the LVLM model.
2. This paper clearly highlights the challenges dealing multimodal compression
compared to single-modal model compression. The modularization of
multimodal models makes compression more challenging, and the paper
provides visualizations of gradients that effectively demonstrate the imbalance
in magnitude and gradient distributions between vision and language models.
3. Experimental results have shown that this method is effective across various
backbones and modal models.

**Weaknesses:**

1. The method in this paper is relatively simple, and novelty is insufficient.
2. The paper's writing could be improved as it lacks a theoretical analysis in the
method introduction section to explain the effectiveness of the proposed method.
Additionally, there is a scarcity of formulas in the paper, and many details are
not adequately clarified.
3. The experiments conducted were not comprehensive enough.
a) The validation of the models in the experiments was also insufficient. Only
the compression effects on the BLIPs model were tested, and the widelyused
CLIP model, which is a mainstream multimodal model, was not
evaluated. Furthermore, the comparison with the UPop model was only
conducted on the NLVR2 and COCO Caption datasets, without
comparisons on other datasets such as Flickr30k and VQA2.0.
b) The ablation study experiments were somewhat simplistic and did not
substantiate the fundamental reasons for the effectiveness of the proposed
method. It would be beneficial to enhance the content of the ablation study
experiments to provide a more detailed analysis and strengthen the overall
credibility of the article.
4. Some minor mistakes:
For example, “and then convert the scores to sparsity by three steps: (1)
Compute the total parameters that need to be selected based on p, (2) Normalize
the scores, (3) Compute the parameters that should be picked for each layer, (4)
Obtain the sparsity for each layer based on the number of parameters to be
picked and the parameters of this layer. ”
The phrase "three steps" in this sentence can also be changed to "four steps."

**Questions:**

See the weakness parts.

---

> ### Author Response · Authors · 2023-11-17
>
> Thank you very much for your review and comments. We provide our response in the following.
> We strongly believe we have clearly addressed all your concerns. Please don’t hesitate to let us know if your concerns/questions are still not clearly resolved. We are open to active discussion and will do our best to address your concerns during the rebuttal period.
>
> ---
>
> > **C1. It lacks a theoretical analysis in the method introduction section to explain the effectiveness of the proposed method. And the method in this paper is relatively simple, and novelty is insufficient.**
>
>
> We respectfully disagree that theoretical analysis is necessary for our introduction since we already provided clear motivations/challenges for tackling efficient pruning for large vision-language models. And, we propose a new method the coarse-to-fine layer-wise pruning approach upon these motivations suggested :
>
> 1) layer-wise one-shot pruning used for large models cannot figure out the optimal pruning rate per layer due to the lack of global (i.e., entire model) information (please see Introduction and Section 3.2).
>
> 2) In large multi-modal models, training modules for different modalities show a significant disparity in their weight and gradient scales. This distributional gap in different modules makes layer-wise one-shot pruning further challenging, and often fails to preserve the model performance with a higher sparsity ratio (please see Figure 1 and Figure 4).
>
>
> From the perspective of novelty, our ECoFLaP takes unique advantages from both layer-wise and global pruning methods. EcoFLaP preserves the performance of full-precision models by computing intra-/inter weight importance (global pruning), yet this process is rapid and efficient (layer-wise pruning) through **our proposed two-step pruning procedures with zeroth-order approximation**. We utilize zeroth-order gradients for global information and bypass the need to construct a backpropagation graph, and thus **our approach can use a similar amount of GPU memory as layer-wise pruning**. To the best of our knowledge,**there has been no previous work similar to our idea for network pruning** and we believe this novel approach sufficiently contributes to the large and/or multimodal model pruning fields with clear insights.
>
> To demonstrate the effectiveness of the proposed method, we provide extensive empirical evidence/analyses with multiple architectures (BLIP2, FlanT5, ViT, LLaMA) and tasks (VQA, Captions, MMLU, ImageNet, WikiText-2, etc), and we also report the results on CLIP in the next response. In those experiments, our approach outperforms the baseline by a margin in the high-sparsity regime, and we believe that **compared to theoretical analysis, the actual experimental results are the most straightforward and useful way to demonstrate the effectiveness of our approach**.
>
> ---
>
> > **C2. there is a scarcity of formulas in the paper, and many details are not adequately clarified.**
>
> We politely disagree with the notion that using many formulas necessarily makes the paper clearer. And we believe we already have included all the essential equations required to explain the algorithm effectively.
> Could you please help specify which parts you think are unclear, so we can address them more clearly/thoroughly in our rebuttal?

---

> > ### Author Response · Authors · 2023-11-21
> > **New results on VQA and a gentle reminder for the rebuttal deadline**
> >
> > Dear reviewer q2nJ:
> >
> >
> > We have finished the VQA experiments on BLIP to compare with UPop more thoroughly. Due to the time constraint, in the fine-tuning experiment of ECoFLaP (the VQA result of the last row), we only used 1/3 of the training data that UPop used. The following table **shows the effectiveness of ECoFLaP even with fewer data**. We are going to add the full fine-tuning results in our revision and we expect the performance would be further improved.
> >
> > Thank you for your constructive feedback on additional tasks (VQA and Flickr30k), and we believe these results further strengthen our paper.
> >
> > | Methods                   | VQA (test dev) | Flickr30k (TR@1/IR@1) |
> > |---------------------------|----------------|-----------------------|
> > | Full model                | 77.4           | 96.8/86.9             |
> > | Wanda (w/o fine-tuning)   | 71.9 (-5.5)           | 85.3/72.3 (-11.5/-14.6)             |
> > | ECoFLaP (w/o fine-tuning) | 73.6 (-3.8)           | 90.2/79.5 (-6.6/-7.4)             |
> > | UPop (w/ fine-tuning)     | 76.3 (-1.1) *           | 94.0/82.0 (-2.8/-4.9)             |
> > | ECoFLaP (w/ fine-tuning)  | **76.7** (-0.7)           | **96.8**/**85.6** (-0.0/-1.3)             |
> >
> > **Use the full training data for fine-tuning*
> >
> > ---
> >
> > **This is also a reminder that tomorrow (Nov 22) is the last day** of the rebuttal, and we would like to follow up to see if the response addresses your concerns or if you have any further questions. We would really appreciate the opportunity to discuss this further if our response has not already addressed your concerns. **If our response addresses your concerns, please kindly consider increasing the scores**. Thank you again!

---

> > > ### Author Response · Authors · 2023-11-23
> > >
> > > Dear reviewer q2nJ,
> > >
> > > Thank you for your valuable time and the constructive feedback you have provided. We also have made efforts to reply to your concerns accordingly. It is only a couple of hours before the rebuttal deadline, and it would be great if you could let us know if our response addresses your concerns or if you have any further questions. Thank you again!

---

> ### Author Response · Authors · 2023-11-17
>
> > **C3. The experiments conducted were not comprehensive enough. a) The validation of the models in the experiments was also insufficient. Only the compression effects on the BLIPs model were tested, and the widelyused CLIP model, which is a mainstream multimodal model, was not evaluated.**
>
> Thank you for your suggestions on more experiments on CLIP. However, as we mentioned in the Introduction and Section 3.1, in the paper, we focused on the modularized vision-language model, in which the vision and language components are trained independently. We also found this kind of modularized VL model (Flamingo, MiniGPT series, LLaVA, Qwen-VL, etc) becomes more common as the strong LLM emerges, and therefore our study will be useful for these VL models. On the other hand, we didn’t conduct experiments on CLIP because its vision and language transformers are trained together, which makes it not in our target models. However, we still perform the experiments on CLIP during the rebuttal, and show the results in the following table. We use 11 classification tasks to evaluate the zero-shot performance of CLIP before and after pruning. We ran Wanda and ECoFLaP over the sparsity 0.3 to 0.5, and we observed that ECoFLaP outperforms Wanda in the three sparsity ratios (we only report the results on sparsity 0.4 to make the reply succinct). In the 0.4 sparsity, **our approach shows a 5% to 18% improvement over Wanda in the 11 tasks**. We have included this experiment in Table 8 and Appendix F of our revision.
>
> | Methods                |  caltech101 | dtd | eurosat | fgvc_aircraft | food101 | imagenet | oxford_flowers | oxford_pets | stanford_cars | sun397 | ucf101 |
> |------------------------|---|---|---|---|---|---|---|---|---|---|---|
> | Full Model             |  92.9 | 44.5 | 47.8 | 24.8 | 86.1 | 66.7 |    71.3  | 89.1 | 65.3 | 62.6 |66.8 |
> | Wanda @ 0.4 sparsity   | 78.9 | 23.9 | 21.1  | 5.9  | 50.0 | 36.8 |    30.4 | 56.0  | 24.6  | 42.9  | 48.2|
> | ECoFLaP @ 0.4 sparsity | **86.6** | **31.7** | **27.1** | **12.9** | **65.0** | **48.7** |   **46.8**  | **74.6**  | **40.1**  | **53.9** | **57.7** |
> ---
>
> > **C4. Furthermore, the comparison with the UPop model was only conducted on the NLVR2 and COCO Caption datasets, without comparisons on other datasets such as Flickr30k and VQA2.0.**
>
> We also ran the Flickr30k experiments to compare with UPop and show the results in the following Table. We show that ECoFLaP outperforms Wanda in the no-fine-tuning setting, and outperforms UPop after the fine-tuning with almost no performance drop. This trend is the same as what we found in the NLVR2 and COCO experiments. We are running the VQA experiments and will release the results soon.
>
>
> | Methods                   | Flickr30k (TR@1/IR@1) |
> |---------------------------|-----------------------|
> | Full model                | 96.8/86.9             |
> | Wanda (w/o fine-tuning)   |  85.3/72.3    (-11.5/-14.6)           |
> | ECoFLaP (w/o fine-tuning) | 90.2/79.5  (-6.6/-7.4)              |
> | UPop (w/ fine-tuning)     | 94.0/82.0  (-2.8/-4.9)             |
> | ECoFLaP (w/ fine-tuning)  |   **96.8**/**85.6** (-0.0/-1.3)            |

---

> ### Author Response · Authors · 2023-11-17
>
> ---
> > **C5. The ablation study experiments were somewhat simplistic and did not substantiate the fundamental reasons for the effectiveness of the proposed method. It would be beneficial to enhance the content of the ablation study experiments to provide a more detailed analysis and strengthen the overall credibility of the article.**
>
> To show the benefits of our main novelty in **“layer-wise pruning with global dynamic sparsity ratios”**, we demonstrated that our proposed approach is stronger than the plain layer-wise pruning methods on various models and tasks in Table 1, Figure 4, and Figure 7. **We believe these experiments have proved the effectiveness of our hypothesis on “dynamic layer ratios are critical for layer-wise pruning”**, and this is the fundamental reason that our approach outperforms the baselines. Moreover, in Table 1, we also show that **our approach uses a lot less GPU memory than other methods using first-order gradients**, and this justifies our idea of replacing first-order gradients with forward-only zeroth-order gradients.
>
> For the ablation studies, we **comprehensively presented the hyperparameter choice of the zeroth-order optimization**, which is the main component of our approach. We showed that the results are quite **robust to the hyperparameters of the zeroth-order optimization**. In the last paragraph of Section 6.2 (ablation study) and Figure 6, we also visualized the loss landscape of BLIP-2 and found the landscape is bell-shaped with a smooth basin and a single local minimum. **This loss landscape also justifies the success of utilizing zeroth-order optimization** as we avoid sampling weights located in other basins that may cause an inaccurate gradient estimation.
>
> In sum, we believe that our experiments have adequately justified the main reason for the effectiveness of our approach and hypothesis. Could you please be more specific about which part of the experiments is unclear or what hypothesis needs to be clarified more, so we try to address them during the rebuttal?
>
> ---
>
> > **C6. Some minor mistakes: For example, The phrase "three steps" in this sentence can also be changed to "four steps."**
>
> Thank you for catching the typo, and we have incorporated it in our revision.
>
> ---
>
> *If our response addresses your concerns, please consider increasing the scores. Also, feel free to ask follow-up questions during the rebuttal period.*

---

### Official Review · Reviewer_5Mvp · 2023-10-31

**Soundness:** 3 good
**Presentation:** 4 excellent
**Contribution:** 4 excellent
**Rating:** 6
**Confidence:** 4

**Summary:**

This paper aims to develop an efficient pruning methods for BLIP like multimodal models. Unlike previous layer-wise pruning methods, this paper proposed a global importance score which can be efficiently approximated using the global model gradients. Layer-wise weight pruning then was applied on the multimodal model. The proposed methods were compared with recently developed pruning methods and consistently improve upon existing methods on accuracy with the same level of sparsity.

**Strengths:**

1. This paper is easy to consume and logically smooth.
2. The scope of this paper is well defined, which is to prune Blip-like multimodal architecture. Challenges of pruning Blip-like models were presented clearly.
3. The key idea of using zeroth-order approximated gradient makes computing the global important score efficient, which is useful.

**Weaknesses:**

1. In addition to the numerical comparison, can you show some example results that compare the before and after results?
2. While it is not easy to measure the real performance improvement, can you discuss about it with the proposed pruning method?
3. Based on the importance score, what are important layers? What's the distribution of scores across all layers?

**Questions:**

Please refer to the weakness section.

---

> ### Author Response · Authors · 2023-11-17
>
> Thank you very much for your review and comments. We provide our response in the following.
> We strongly believe we have clearly addressed all your concerns. Please don’t hesitate to let us know if your concerns/questions are still not clearly resolved. We are open to active discussion and will do our best to address your concerns during the rebuttal period.
>
> ---
>
> > **C1. In addition to the numerical comparison, can you show some example results that compare the before and after results?**
>
> We conducted an analysis comparing the predictions of the Full model (before pruning) and the model pruned by ECoFlaP (with 0.5 sparsity) on the GQA datasets, and presented three illustrative examples in Table 9 (in Appendix G) of our revision. Please refer to them for the details.
>
> We found one interesting observation: out of 12578 questions, the Full model correctly answered 1219 questions that our model did not, yet our model also accurately responded to 1072 questions where the Full model failed to respond. **This indicates that pruning does not always lead to performance degradation; in some instances, it may even enhance model robustness for certain questions** [1]. During our review of incorrect predictions by either our model or the Full model, we noticed that most were conceptually similar to the ground truth. This suggests that the pruned model still retains most of the capability. Furthermore, in some cases, such as with the 'horses running' example, both models provided answers that might be considered correct by many humans, even though they differ from the ground truth.
>
> [1] Zhangheng Li, Tianlong Chen, Linyi Li, Bo Li, Zhangyang Wang, Can Pruning Improve Certified Robustness of Neural Networks?
>
> ---
>
> > **C2. While it is not easy to measure the real performance improvement, can you discuss about it with the proposed pruning method?**
>
> In this paper, we evaluate the pruning approaches on large pre-trained models (BLIP2, FlanT5, ViT, LLaMA) with downstream tasks in a zero-shot setting, and this approach is one of the most common ways to benchmark pre-trained models. With this evaluation approach, we demonstrated that ECoFLaP has an improvement over the compared approaches on various architectures and downstream tasks in Table 1, Figure 4, and Figure 7. While we have tried our best to evaluate the models in this way, could you elaborate more on the method in your mind that can measure the real performance improvement, so we can conduct the method during the rebuttal phase?
>
> ---
>
> > **C3. Based on the importance score, what are important layers? What's the distribution of scores across all layers?**
>
> Our sparsity ratios for layers are computed by the importance scores and thus the ratios can reflect the importance, for example, the higher sparsity for one layer denotes that this layer is less important for the tasks. In Figure 5, we can see that the sparsity ratio generally is lower for the vision transformer and higher for the language model, and this suggests that the language model is more prunable and has more redundancy for the vision-language tasks.
>
> ---
>
> *If our response addresses your concerns, please consider increasing the scores. Also, feel free to ask follow-up questions during the rebuttal period.*

---

### Official Review · Reviewer_GD72 · 2023-11-01

**Soundness:** 3 good
**Presentation:** 3 good
**Contribution:** 2 fair
**Rating:** 6
**Confidence:** 4

**Summary:**

The paper addresses the challenges of deploying Large Vision-Language Models (LVLMs) due to their high computational and energy costs. Traditional global pruning methods are costly, and recent layer-wise pruning approaches lack a global perspective. To overcome this, the paper introduces Efficient Coarse-to-Fine Layer-Wise Pruning (ECoFLaP), a two-stage approach that leverages global importance scores for determining sparsity ratios and then performs efficient layer-wise weight pruning. The computation of global importance scores is achieved using the zeroth-order approximate gradient through a forward-forward algorithm.   ECoFLaP demonstrates significant performance improvements in both multimodal and single-modal models, particularly in high-sparsity scenarios. Notably, it achieves these improvements while using only 40% of GPU memory compared to backpropagation and maintaining competitive accuracy.

**Strengths:**

+ Global pruning indeed faces challenges in sparsity allocation, particularly for multi-modal models. The paper provides a thorough and persuasive analysis of the motivation behind this issue.
+ The method presented in the paper is straightforward yet proven to be effective, and the experiments conducted are comprehensive.
+ The manuscript is readily understandable and offers sufficient experimental details for reproducibility.

**Weaknesses:**

- There is a need for a more detailed description of Algorithm 1.
- Several metrics are used for importance scores, including weight magnitude, the multiplication of gradient and weight magnitude, gradient only, and sensitivity analysis. It would be beneficial to provide a comprehensive comparison of these metrics.
- There are some typos in this paper: (1) In Section 4.1, "and then convert the scores to sparsity by three steps" should be "and then convert the scores to sparsity by four steps"? (2) In Section 4.2, "like less than 5B parameters" should be "Like less than 5B parameters"?

**Questions:**

- The sparsity ratios in the experiments cover a range from 0.1 to 0.6. Is it possible to extend this range to include a wider spectrum of sparsity ratios?
- Can this method be combined with SparseGPT?

---

> ### Author Response · Authors · 2023-11-17
>
> Thank you very much for your review and comments. We provide our response in the following.
> We strongly believe we have clearly addressed all your concerns. Please don’t hesitate to let us know if your concerns/questions are still not clearly resolved. We are open to active discussion and will do our best to address your concerns during the rebuttal period.
>
> ---
>
> > **C1. There is a need for a more detailed description of Algorithm 1.**
>
> Sorry to hear that you feel further details are needed in Algorithm 1. However, we made effort to providing  extensive explanations of our algorithm using comments in Algorithm 1 and also used the equations in Section 4.1 as support. Could you please elaborate more detail which part is unclear so we can improve it during the rebuttal phase?
>
>
> ---
>
> > **C2. Several metrics are used for importance scores, including weight magnitude, the multiplication of gradient and weight magnitude, gradient only, and sensitivity analysis. It would be beneficial to provide a comprehensive comparison of these metrics.**
>
> Thank you for your suggestion. We presented a table to compare the approaches in Table 1 in terms of whether they are global or not and the used importance measure. We have included this table in our revision (Please see Table 6 in the Appendix).
>
> | Methods                  | Global or Layer-wise | Importance Measure                                                                |
> |--------------------------|----------------------|-----------------------------------------------------------------------------------|
> | Global Magnitude Pruning | Global               | $\| W \|_{ij}$                                                                   |
> | Iterative OBD Pruning    | Global               | $ \| W \| _{ij} \cdot \| dL / dW \| _{ij}$                                               |
> | SparseGPT                | Layer-wise           | $[\|W\|^ 2 / diag (XX^T + λI)^{−1}]_{ij}$                                         |
> | Wanda                    | Layer-wise           | $\|W_{ij}\| \cdot \|X_{j}\|^2 $                                                   |
> | ECoFLaP (zeroth-order)   | Mixed     | First Stage: $ \| dL / dW \| _{ij} $ Second Stage: $\|W _{ij}\| \cdot \|X _{j}\|^2 $ |
>
> ---
>
> > **C3. There are some typos in this paper: (1) In Section 4.1, "and then convert the scores to sparsity by three steps" should be "and then convert the scores to sparsity by four steps"? (2) In Section 4.2, "like less than 5B parameters" should be "Like less than 5B parameters"?**
>
> Thank you for catching the typos. We have incorporated them in the revision.
>
> ---
>
> > **C4. The sparsity ratios in the experiments cover a range from 0.1 to 0.6. Is it possible to extend this range to include a wider spectrum of sparsity ratios?**
>
> Thank you for your suggestion on evaluation with a wider sparsity range. During the rebuttal period, we have compared Wanda and ECoFLaP at 0.7 sparsity in the following table.
>
> As shown, ECoFLaP consistently outperforms Wanda by a significant margin, especially on NoCaps and Flickr30k (30% up improvement). This is because recent layer-wise pruning approaches can only compute local (i.e., within-layer) importance, which leads to suboptimal compression, while our proposed method can efficiently and accurately prune model weights based on estimated global importance. In this case, our approach **assigns lower pruning ratios on the vision transformer, and this prevents a serious performance drop at such a high sparsity**.
>
> | **Methods @ 0.7 sparsity** | **VQA (Acc)** | **OK-VQA (Acc)** | **GQA (Acc)** | **NoCaps (CIDEr/SPICE)** | **Flickr30k (TR@1/IR@1)** |
> |--------------------|---------------|------------------|---------------|--------------------------|---------------------------|
> | Wanda                      | **4.76**          | 3.76             | 2.96          |  8.13/5.97         | 7.6/22.76                 |
> | ECoFLaP (zeroth-order)     | **4.76**          | **5.05**             | **4.14**          | **38.28**/**8.74**         | **78.1**/**68.34**      |

---

> ### Author Response · Authors · 2023-11-17
>
> > **C5. Can this method be combined with SparseGPT?**
>
> Thank you for your constructive suggestion. As the key contribution of our method for efficient estimation of layer-adaptive pruning rate for layer-wise pruning is orthogonal to SparseGPT, ECoFLaP can also be combined with SparseGPT. As shown in the table below, EcoFLaP+SparseGPT consistently achieves superior performance over multiple tasks.
>
> Additionally, the performance improvement seems relatively smaller compared to ECoFLaP with Wanda (i.e., original EcoFLaP). We hypothesize that this is because the reweighting mechanism in SparseGPT is somewhat correlated to our dynamic sparsity ratios, where both of them are important to preserve the performance in high sparsity ratios. We have added this discussion and results in Appendix E and Table 7 of our revision.
>
> | **Methods @ 0.6 sparsity** | **VQA (Acc)** | **OK-VQA (Acc)** | **GQA (Acc)** | **NoCaps (CIDEr/SPICE)** | **Flickr30k (TR@1/IR@1)** |
> |--------------------|---------------|------------------|---------------|--------------------------|---------------------------|
> | SparseGPT    | 48.68  | 27.88      | 35.10       |  101.04/13.41       | 95.2/84.98 |
> | ECoFLaP on SparseGPT (zeroth-order)     | **50.36**          | **27.93**             | **36.26**          | **105.25**/**13.96**   | **96.3**/**86.12**      |
>
> ---
>
> *If our response addresses your concerns, please consider increasing the scores. Also, feel free to ask follow-up questions during the rebuttal period.*

---

> ### Author Response · Authors · 2023-11-21
> **Gentle reminder of the rebuttal deadline**
>
> Dear reviewer GD72:
>
> This is a reminder that tomorrow (Nov 22) is the last day of the rebuttal, and we would like to follow up to see if the response addresses your concerns or if you have any further questions. We would really appreciate the opportunity to discuss this further if our response has not already addressed your concerns. If our response addresses your concerns, please kindly consider increasing the scores. Thank you again!

---

### Official Review · Reviewer_4r21 · 2023-11-04

**Soundness:** 2 fair
**Presentation:** 3 good
**Contribution:** 2 fair
**Rating:** 5
**Confidence:** 4

**Summary:**

This paper presents the Efficient Coarse-to-Fine Layer-Wise Pruning (ECoFLaP) method for Large Vision-Language Models. The method utilizes zeroth-gradient optimization to determine layer-specific sparsity ratios and applies layer-wise unstructured weight pruning. Experimental results demonstrate improvements on some benchmarks.

**Strengths:**

-- Pruning is a crucial topic for efficient large models.

-- The proposed method is interesting to me.

-- Extensive experimental results demonstrate some improvement compared to previous methods.

**Weaknesses:**

Overall, the paper leans more towards an engineering-focused approach. The proposed method appears to be a straightforward combination of existing techniques without offering mathematical insights or clear motivations. The paper requires significant revisions to improve its writing quality.

-- The paper claims that pruning multi-modal large models differs from other large models, but lacks mathematical or experimental support for this assertion.

-- The use of layer-wise pruning due to the computational cost of calculating the inverse of the Hessian is common in other large model pruning approaches.

-- The assumption of having sufficient GPU resources in the pruning scenario is not adequately discussed.

-- While zeroth-gradient optimization is effective for efficient fine-tuning of large models [1], its motivation and suitability in a pruning scenario, considering challenges like slow convergence and sensitive hyper-parameters, remain unclear.

-- The use of calibration datasets in experiments raises concerns about fairness and the significance of employing zeroth-order optimization to save memory if calibration datasets are used for fine-tuning.

The paper suffers from poor writing quality and contains noticeable typos:

1. that finds adaptive sparsity per layer by leveraging the global importance score approximated via first-order gradients. (should be zeroth-order gradients)

2. Note that our proposed method is computationally efficient by leveraging the first-order gradient to obtain a global importance score without Hessian operations (should be zeroth-order gradients)

[1] Malladi, Sadhika, et al. "Fine-Tuning Language Models with Just Forward Passes." arXiv preprint arXiv:2305.17333 (2023).

**Questions:**

Please refer to my detailed comments in the weakness part.

---

> ### Author Response · Authors · 2023-11-17
>
> Thank you very much for your review and comments. We provide our response in the following.
> We strongly believe we have clearly addressed all your concerns. Please don’t hesitate to let us know if your concerns/questions are still not clearly resolved. We are open to active discussion and will do our best to address your concerns during the rebuttal period.
>
> ---
>
> >  **C1. The proposed method appears to be a straightforward combination of existing techniques without offering mathematical insights or clear motivations.**
>
>
> We want to emphasize that we already provide two clear motivations for our approach:
>
> 1) Layer-wise one-shot pruning used for large models cannot figure out the optimal pruning rate per layer due to the lack of global (i.e., entire model) information (please see Introduction and Section 3.2).
>
> 2) In large multi-modal models, training modules for different modalities show a significant disparity in their weight and gradient scales. This distributional gap in different modules makes layer-wise one-shot pruning further challenging, and often fails to preserve the model performance with a higher sparsity ratio (please see Figure 1 and Figure 4).
>
> To address these two key problems in pruning large multi-modal models, we propose our method, ECoFLaP.
>
> Our design takes strong advantages from both layer-wise and global pruning methods. Our EcoFLaP preserves the performance of full-precision models by computing intra-/inter weight importance (global pruning), yet this process is rapid and efficient (layer-wise pruning) through our proposed two-step pruning procedures with zeroth-order approximation. We utilize **zeroth-order gradients for global information and bypass the need to construct a backpropagation graph**, and thus **our approach can use a similar amount of GPU memory as layer-wise pruning**. To the best of our knowledge, we didn’t see previous work utilizing our idea for network pruning.
>
>
>
> We argue that our approach **provides insights into both layer-wise and global pruning approaches** and **the method is practical to use a minimal amount of GPU memory**. We also presented our motivations in the Introduction and Section 3.2.
>
> ---
>
> > **C2.The paper claims that pruning multi-modal large models differs from other large models, but lacks mathematical or experimental support for this assertion.**
>
> We politely argue that this is a misunderstanding. we clearly provided analyses and results that pruning multi-modal models differs from pruning single-modal models in Section 3.2, and **conventional single-modal layer-wise pruning methods suffer from significant performance degeneration in multi-modal model in high sparsity regime (Please see Figure 4(a))**. In Section 3.2 and Figure 1(c), we demonstrated that **local score estimation is imbalanced for vision and language models**, which causes inaccurate estimation of the pruning ratios for each layer in the model. Also, the local score of each parameter cannot infer the importance of the final loss, and that is one reason that layer-wise pruning only uses a fixed pruning ratio for layers.
>
> ---
>
> > **C3. The use of layer-wise pruning due to the computational cost of calculating the inverse of the Hessian is common in other large model pruning approaches.**
>
> We believe there is a misunderstanding in our contribution.
> Our contribution is not on layer-wise pruning itself. As described in the second paragraph of Section 3.2, we clarify the critical challenges in pruning large vision-language multi-modal models, and suggest a simple yet powerful pruning approach using the **memory-efficient zeroth-order optimization to estimate dynamic pruning ratio per layer with minimal computations**. Our EcoFLaP successfully mitigates severe performance drops even with a high degree of model compression (Figure 4), by overcoming the limitations of general layer-wise pruning methods.
>
> ---
>
> > **C4. The assumption of having sufficient GPU resources in the pruning scenario is not adequately discussed.**
>
> In Table 1, we showed that ECoFLaP only uses slightly more GPU memory (0.06GB) than Wanda and even less memory than SparseGPT. Furthermore, ECoFLaP also only uses 0.5GB more memory than Magnitude Pruning (which directly uses weight magnitude as the importance score and does not require any computation). We believe that the efficiency of ECoFLaP makes it a strong method as layer-wise pruning in the low computational cost regime.

---

> ### Author Response · Authors · 2023-11-17
>
> > **C5. While zeroth-gradient optimization is effective for efficient fine-tuning of large models [1], its motivation and suitability in a pruning scenario, considering challenges like slow convergence and sensitive hyper-parameters, remain unclear.**
>
> We believe there is a clear misunderstanding in our pruning strategy. We note that our proposed method is a **one-shot pruning approach, and there is no convergence issue as we didn’t use zeroth-order optimization to update the network like [1]**.  We only use zeroth-order optimization to **estimate the gradient** of the pre-trained weight **once** and use the gradient as the importance of parameters. Moreover, our ablation studies in Tables 3 - 5 demonstrated that our approach is robust to hyperparameters of zeroth-order optimization.
>
> ---
>
> > **C6. The use of calibration datasets in experiments raises concerns about fairness and the significance of employing zeroth-order optimization to save memory if calibration datasets are used for fine-tuning.**
>
> There is a clear misunderstanding of the basics of one-shot pruning. We first note that using small calibration data is a standard approach for one-shot (layer-wise) pruning [2,3,4]. And we also followed the same setting for our approach and didn’t re-train the model using the calibration set after pruning.
>
> [2]  Hubara, I., Chmiel, B., Island, M., Banner, R., Naor, S., and Soudry, D. Accelerated sparse neural training: A provable and efficient method to find N:M transposable masks.
>
> [3] Elias Frantar, Dan Alistarh, SparseGPT: Massive Language Models Can be Accurately Pruned in One-Shot
>
> [4] Mingjie Sun, Zhuang Liu, Anna Bair, J. Zico Kolter, A Simple and Effective Pruning Approach for Large Language Models
>
> ---
>
> > **C7. that finds adaptive sparsity per layer by leveraging the global importance score approximated via first-order gradients. (should be zeroth-order gradients)**
>
> Thanks for catching the typo. We have reflected this in the revision.
>
> ---
>
> > **C8. Note that our proposed method is computationally efficient by leveraging the first-order gradient to obtain a global importance score without Hessian operations (should be zeroth-order gradients)**
>
> This part is not a typo. We first show another version of our approach with the first-order gradient (we also show the result of this version in Table 1), which is cheaper than the second-order Hessian method, and then propose a more efficient version with the zeroth-order gradient.
>
> ---
>
> *If our response addresses your concerns, please consider increasing the scores. Also, feel free to ask follow-up questions during the rebuttal period.*

---

> > ### Comment · Reviewer_4r21 · 2023-11-21
> >
> > Thank you for your detailed response and the effort put into addressing the concerns raised. After careful consideration of your responses and a re-evaluation of the manuscript, I have decided to maintain my original score.

---

> > > ### Author Response · Authors · 2023-11-21
> > > **We are keen to further discuss any unaddressed issues**
> > >
> > > Dear reviewer 4r21,
> > >
> > > We are sincerely grateful to you for reading our response and we are glad that you appreciate our **effort put into addressing the concerns**. It would be great if you could give us **further insights/unaddressed issues** or kindly consider revising your score of "3: reject"?
> > >
> > > During the rebuttal period,
> > > - We have emphasized that **our proposed method is based on strong and clear motivations that multi-modal large models differ from other large models**, including **empirical evidence**.
> > > - To provide more empirical evidence, we also conduct additional experiments (please refer to our reply to Reviewer GD72) on CLIP models and two more datasets on BLIP to compare with UPop (VQA and Flickr30k). We demonstrate that our approach shows a **5% to 18% improvement over Wanda in the 11 tasks with CLIP**, and also **outperforms UPop on VQA and Flickr30k**.
> > > - We have explained the **misunderstanding regarding our contribution**; our contribution is not to introduce layer-wise pruning.
> > > - We have explained the misunderstanding regarding the convergence of our method; our proposed method is a one-shot pruning approach, and **there is no convergence issue as we didn't use zeroth-order optimization to update the network**.
> > > - We have explained the misunderstanding regarding the one-shot pruning; **using small calibration data is a standard approach for one-shot (layer-wise) pruning**.
> > >
> > > We remain committed to further improving the quality of our paper by addressing any remaining concerns and suggestions where necessary. With that in mind, please let us know if you have any further feedback. We would be grateful for the opportunity to address them and make our work a more solid and valuable contribution to the field of large-scale multimodal model pruning.
> > >
> > > Also, we would like to **kindly suggest reconsideration of the rating, if you feel that our work does not have major concerns for evaluation, resources, reproducibility, and ethical considerations**. We understand that the criteria for rating a paper can sometimes be subjective; however, we believe that most of your concerns are effectively addressed as long as there are no significant issues.
> > >
> > > We thank you so much for your time and effort in reviewing our paper and for the constructive feedback that has greatly improved it.
> > >
> > > Warm Regards,
> > > Authors

---

> > > ### Comment · Reviewer_4r21 · 2023-11-22
> > > **Additional Comment**
> > >
> > > I genuinely value the authors' efforts and the additional results they have incorporated.
> > >
> > > Upon thorough examination, I acknowledge the insights presented in this paper concerning the challenges associated with pruning the vision-language model (referenced in Section 3.2). The statistics regarding weight magnitudes, gradients, and SparseGPT's local score highlight a disparity between the visual and linguistic components of the Vision Language Model (VLM). Moreover, as noted in Section 3.2, the prevalent layer-wise pruning methods overlook the differential nature of the vision and language models, an observation that I find particularly innovative.
> > >
> > > However, I maintain significant reservations regarding two critical aspects:
> > >
> > > 1. The paper asserts distinct differences between the vision and language segments within the VLM, yet the applied scoring function (Equation 5) remains uniform for both. Given that this function is intended to assess the relevance of each layer in the VLM, **the paper seemingly contradicts its own rationale by not differentiating between the two segments**.
> > >
> > > 2. Concerning the primary technical contribution of the paper detailed in Section 4.2, it employs zeroth-order gradient approximation for a layer's weight to circumvent the substantial memory demands associated with storing first-order gradients. Nonetheless, Equation 5 implies that computing the score (L2 norm of zeroth-order gradients) for layer i necessitates a memory allocation identical to that needed for storing first-order layer-wise gradients. **This suggests that zeroth-order gradient use does not yield memory savings when compared with first-order gradients.** This claim (utilizing zeroth-order gradients save memory) is further challenged by the experimental data in Table 1, where using zeroth-order gradients for the BLIP-2 model at 0.5 sparsity consumes 8.93GB of GPU memory, comparable to other layer-wise methods approximating 9GB. Conversely, the ECoFLaP + First-order Gradient approach incurs a 22.4GB memory cost, matching that of Iterative OBD Pruning, which accounts the entire model's first-order gradient and weight magnitude multiplication. These findings imply that ECoFLaP + First-order Gradient's reported memory usage encompasses the entire model's 1st-order gradients. Hence, a more equitable memory usage comparison between layer-wise zeroth and first-order gradients is warranted, potentially leading to analogous outcomes.
> > >
> > > In summation, while the paper sheds new light on the difference between the visual and linguistic components in VLMs, it fails to bride the gap. The main technical proposition of the paper—the use of layer-wise zeroth-order gradient norms as a scoring mechanism—counters the authors' stated objectives of solving the difference between VLM components and achieving memory efficiency. Therefore, my assessment remains unchanged, and I lean towards rejecting this paper.

---

> > > > ### Author Response · Authors · 2023-11-22
> > > > **Clarification on additional questions**
> > > >
> > > > Thank you for your detailed feedback! We are glad that you “*acknowledge the insights presented in this paper concerning the challenges associated with pruning the vision-language model*” and find it “*particularly innovative*”. We provide further clear explanations for your new comments and we believe they can address your concerns and you could revisit your score accordingly.
> > > >
> > > > ---
> > > >
> > > > > **The paper asserts distinct differences between the vision and language segments within the VLM, yet the applied scoring function (Equation 5) remains uniform for both**
> > > >
> > > > The goal of our algorithm is to determine the importance of each layer in different modules (vision and language transformers) and then convert the importance to sparsity ratio for layer-wise pruning. Our intuition is that the layer should be more important if one layer can cause a larger “**loss change of final output**” (e.g. global loss), and this idea leads us to use the global gradients of each layer, which is Equation (5), to estimate the importance score of different layers in different modules. **Even though we use the same equation for vision and language models, their importance scores will be different because the impact on the global loss of each layer in the two modules is different**.
> > > >
> > > > In the above paragraph, we show that our design can differentiate the importance of different modules with one equation, and hence the motivations and the proposed approach **do not have any contradictions regarding rationale** that the reviewer is worried about.
> > > >
> > > > ---
> > > >
> > > > > **Nonetheless, Equation 5 implies that computing the score (L2 norm of zeroth-order gradients) for layer i necessitates a memory allocation identical to that needed for storing first-order layer-wise gradients. This suggests that zeroth-order gradient use does not yield memory savings when compared with first-order gradients.**
> > > >
> > > > Thank you for your interesting questions. **There might be a misunderstanding and we would like to clarify that the zeroth-order optimization algorithm [1] is extremely memory-efficient and doesn’t have to use a lot of extra memory**. This approach modifies the weights “**in-place**” so that we don’t need a lot of extra memory to store the perturbed weights and the gradient norm (is a scalar) in Equation (5). The exact process for Equation (5) is as follows, for layer i:
> > > >
> > > > * First, we choose a random seed and then sample a noise $z$ (now we use 2x of memory of “this layer”).
> > > > * We modify the model’s weight with $W+ \epsilon z$ ($\epsilon$ is a constant) and compute the loss change $\mathcal{L}(W+ \epsilon z, d)$.
> > > > * We choose the same random seed as the first step and then sample the same noise $z$.
> > > > * We modify the model’s weight with $W- \epsilon z$, which can be done by subtracting $2 \epsilon z$ from the previous weight, and compute the loss change $\mathcal{L}(W - \epsilon z, d)$
> > > > * Compute the gradient norm by ($\mathcal{L}(W + \epsilon z) - \mathcal{L}(W - \epsilon z)) / (2\epsilon)$ (note that all terms are scalars)
> > > >
> > > >
> > > > From the above process, for an N-layer network, **we only need (N+1)x of one-layer memory, which is much more memory-efficient than storing the whole gradients that cause 2N of one-layer memory**. This explains why our approach uses a lot less memory than first-order based methods (these methods also need to cache input activations), uses similar memory as layer-wise methods (which use extra memory to compute layer scores), and uses only 0.5GB more GPU memory compared to Magnitude Pruning.
> > > >
> > > > We could not expand the whole process that happens in Equation (5) in the original paper due to the page limit but we will add this in the revised pdf (coming later today).
> > > >
> > > >
> > > > [1] Sadhika Malladi, Tianyu Gao, Eshaan Nichani, Alex Damian, Jason D. Lee, Danqi Chen, Sanjeev Arora, “Fine-Tuning Language Models with Just Forward Passes”, 2023

---

> > > > > ### Author Response · Authors · 2023-11-22
> > > > > **The revised pdf is uploaded**
> > > > >
> > > > > We just uploaded the revised PDF. On page 6, we expand the process of zeroth-order optimization and explain its memory efficiency (new content is marked as brown).

---

> > > > > > ### Comment · Reviewer_4r21 · 2023-11-23
> > > > > >
> > > > > > I sincerely appreciate for author’s effort in their timely response.
> > > > > >
> > > > > > I accept their clarification on the memory cost of zeroth-order gradients. However, I still kindly remind the memory cost of computing first-order gradients should be carefully considered, since a lower memory cost ( much less than 2N shown in the paper) can be achieved via some back propagation trick.
> > > > > >
> > > > > > Based on the above reasons, I raised my score to 5. I still hold concerns about the consistency of methodology and motivation, but I don’t mind the paper getting accepted.

---

> ### Author Response · Authors · 2023-11-23
>
> Thank you very much for your time to engage in the discussion and for your acknowledgment of our explanation. We will carefully consider your additional feedback and incorporate it into the paper.
>
> Happy Thanksgiving!

---

### Author Response · Authors · 2023-11-17

Thank all the reviewers for the thoughtful reviews. We are encouraged that the reviewers found our approach **interesting** [4r21], **effective** [GD72, 5Mvp, q2nj], and **reproducible** [GD72]. We are also glad to receive reviewers’ positive comments on our **extensive empirical experiments** [4r21, GD72], the **paper’s smooth presentations** [GD72, 5Mvp], and the **clear motivations** [GD72, 5Mvp, q2nj]. In the following, we hope our responses can address your concerns.

We have also submitted a revision based on your valuable feedback (all the updated contents are marked in brown), where we included:
1. Table 6 in the Appendix compares different methods based on their category and used importance measure [GD72].
2. Table 7 in the Appendix compares ECoFLaP with SparseGPT [GD72], and we show that the idea of **ECoFLaP can improve SparseGPT** too.
3. Table 8 in the Appendix compares ECoFLaP with Wanda and the Full Model with the CLIP model [q2nJ], and we demonstrate that **our approach shows a 5% to 18% improvement over Wanda in the 11 tasks**.
4. Table 9 in the Appendix shows qualitative examples to compare the predictions generated by the Full Model and the ECoFLaP-pruned model [5Mvp].
5. Typos are fixed [4r21, q2nJ].

---

### Meta-Review · Area_Chair_zrsT · 2023-12-04

**Metareview:**

This paper focuses on obtaining efficient large vision-language models. Global pruning methods are computationally costly. Therefore, local methods are preferred.
This paper uses a two-stage approach for pruning. First, it uses zeroth-gradient optimization to calculate the global importance scores to determine layer-wise sparsity ratios. Second, local layer-wise unstructured pruning was applied.

**Justification For Why Not Higher Score:**

- Lack of mathematical insights.
- A detailed comparison between different metrics can further improve the paper.
- The novelty seems slightly limited considering numerous previous works in this area, including SparseGPT and Wanda.

**Justification For Why Not Lower Score:**

- Solid empirical evaluations.
- Great rebuttal results.

---

### Decision · Program_Chairs · 2024-01-16

Accept (poster)